# Assessment of the Compliance of Cystitis Management According to French Recommendations through the Analysis of Prescriptions Collected in Community Pharmacies

**DOI:** 10.3390/antibiotics11070976

**Published:** 2022-07-20

**Authors:** Arthur Piraux, Ramy Hammoud, Jérémie Riou, Souhil Lebdai, Sébastien Faure

**Affiliations:** 1University of Angers, Inserm, CNRS, MINT, SFR ICAT, F-49000 Angers, France; sebastien.faure@univ-angers.fr; 2Methodology and Biostatistics Department, University of Angers, F-49000 Angers, France; rahammoud@etud.univ-angers.fr; 3University of Angers, Inserm, CHU Angers, CNRS, MINT, SFR ICAT, F-49000 Angers, France; jeremie.riou@univ-angers.fr; 4Department of Urology, Angers University Hospital, F-49000 Angers, France; solebdai@chu-angers.fr

**Keywords:** urinary tract infection, cystitis management, guideline adherence, antibiotics stewardship, antibiotics, antimicrobial resistance

## Abstract

Urinary tract infections, especially cystitis, are common infections; they are the second most prevalent cause of antibiotic prescriptions in community pharmacies. To reduce antimicrobial resistance, guidelines are revised regularly. This study aims to assess compliance between prescriptions collected in community pharmacies and French cystitis guidelines. A treatment is considered compliant if the nature, dosage, and duration of the antibiotics are correct. Only women aged 18–65 years with a diagnosis of cystitis were eligible. The participation of 16 pharmacies resulted in 303 prescriptions. Most infections were classified as uncomplicated cystitis (79.2%), general practitioners were the prescribers in more than 9 out of 10 cases, and fosfomycin trometamol was the antibiotic dispensed for 1 in 2 women. An average compliance of 66% was observed, but with disparities according to the type of cystitis. Two-thirds of cases of uncomplicated cystitis and recurrent cystitis followed the recommendations, whereas only 15% of cystitis cases that were at risk of complication did so. The inclusion of a urine examination in uncomplicated cystitis decreased the overall compliance rate to 5.8%. These results show the essential role played by pharmacists; they are the last line of defence before dispensing antibiotics. They must know the recommendations in order to apply them.

## 1. Introduction

Urinary tract infections (UTIs) are the second most common cause of antibiotic prescriptions after respiratory infections [1,2,3,4]. Each year, 150 million people are treated for this condition [5]. UTIs include several conditions, such as cystitis, pyelonephritis, urethritis, and prostatitis [6,7]. Cystitis in women is characterized by an inflammation of the bladder caused by germs such as *Escherichia coli* (*E. coli*). It is a commensal pathogen, naturally found in the urinary tract, responsible for 75–95% of infections [6]. The imbalance of its environment can cause its proliferation and lead to an infection [8]. *E. coli* is the most frequently found bacteria. Symptoms such as pollakiuria (urinary frequency) with an urgent need to urinate, dysuria (burning and pain during urination), and hematuria (blood in urine) are frequently observed [8,9].

Three different types of cystitis are described in [3,7,10]. Simple or uncomplicated cystitis is the most common diagnosis, when there are no identified risk factors. Cystitis at risk of complication is diagnosed when a patient presents with at least one risk factor. The most common risk factors are urinary tract abnormality, pregnancy, being elderly (≥75 years old), severe chronic renal failure, and severe immunodeficiency [9,10]. Recurrent cystitis occurs when a patient has at least four episodes of cystitis in the same year. This definition has evolved recently. UTIs are considered recurrent if they occur at least three times a year or twice within the last six months [11].

A urine dipstick is the only test recommendedfor uncomplicated cystitis, but a urine culture is needed for the two other types of cystitis [10,12]. The negative predictive value of a urine dipstick is very high (>95%). It allows the detection of two compounds: leukocytes (evidence of infection) and nitrites (presence of an enterobacteria with a nitrate reductase, such as *E. coli*, *Proteus mirabilis*, or *Klebsiella* spp.). In the case of recurrent cystitis, urine cultures are only requested during the initial episode.

The high frequency of this infection results in the regular consumption of antibiotics, sometimes in the same patient. In France, the choice of the antibiotic is guided by three parameters: efficacity, tolerability, and the ecological impact on the gut microbiota [10]. The most recent studies have shown that a shorter duration of antibiotics improves the above parameters, without limiting the treatment of the infection [13]. These results explain why the duration of the antibiotics to treat a simple acute cystitis is 3 days, and no longer 5 days, with pivmecillinam, or with a single dose of fosfomycin trometamol [9,14]. To reduce the antimicrobial resistance (AMR), the nature of the antibiotic must be considered, and guidelines must be reviewed regularly [15]. For example, the overuse of fluoroquinolones created a resistance to these drugs in uropathogenic *E. coli*. This resistance is particularly marked in developed countries (55–85% resistance), compared with developing countries (5–30% resistance) [16].

A precedent study analysed the impact of new recommendations on antibiotic-prescribing behaviour in France [17]. Several antibiotics are still commonly prescribed, despite their withdrawal from recommendations (lomefloxacin and norfloxacin, in particular). This reflects the length of time it takes for recommendations to reach prescribers. This observation is confirmed by other studies. Hecker et al. report low adherence to UTI guidelines in the emergency department (42%), which led to a stewardship intervention [18]. Adherence to the guidelines increased to 68% (period 1) and 82% (period 2). Grave et al. reported similar adherence to French guidelines for UTIs (66%) [19]. This rate varied significantly depending on the condition; 66% (n = 71) of acute cystitis cases were conformed, but only 20% (n = 3) of cystitis at risk of complication did so. In line with these results, this study aimed to assess compliance between prescriptions collected in community pharmacies and respective French cystitis guidelines.

## 2. Materials and Methods

The development of this study required the completion of a preliminary study to determine the necessary sample size, and to verify its feasibility. Its goal was to propose an overview of the situation to obtain an estimation of the compliance between the treatment and the French guidelines to treat cystitis.

### 2.1. Preliminary Study

A preliminary study was conducted in four community pharmacies to obtain approximately 50 response forms. Each pharmacist was trained prior to the data collection to ensure a proper understanding of the form (Appendix A). It was available online, on the LimeSurvey^®^ platform, and on paper. Before completing the form, it was necessary to check the following inclusion/non-inclusion criteria. Only women aged 18–65 years with a diagnosis of cystitis (uncomplicated cystitis, cystitis at risk of complication, or recurrent cystitis) were eligible. In addition, the lack of pregnancy or another UTI (i.e., pyelonephritis) was required. Each incomplete form was considered non-responsive.

This preliminary study aimed to evaluate the accordance between the guidelines and the treatment prescribed for a UTI. The prescription was considered compliant if, and only if, the nature, the dosage, and the duration of the antibiotics were correct. Current recommendations at the time of the study were used as a reference; they were published by the French Infectious Diseases Society (*Société de Pathologie Infectieuse de Langue Française—SPILF*) in 2018 [10]. Specific guidelines were edited for each type of cystitis (Table 1).

The pilot study was conducted from 22 September to 18 October 2020. In total, 56 forms were collected but only 52 were retained (1 patient was pregnant, 1 patient was older than 65 years, and there were 2 incomplete forms). A total of 41 prescriptions were in compliance with the recommendations regarding the nature of the antibiotic. However, only 36 prescriptions were fully compliant with national recommendations, representing an observed compliance percentage of 69.2%.

### 2.2. Main Study

The main study was conducted from 14 January to 1 June 2021. All data had to be entered into the LimeSurvey^®^ platform, but pharmacists had the possibility to use a paper questionnaire with the patient before completing the online questionnaire. The inclusion and non-inclusion criteria were the same as those used in the pilot study (women, aged between 18 and 65 years old, a diagnosis of cystitis, no pregnancy, and no other UTIs present).

The prescription of antibiotics in a woman was a sign that the patient would require the pharmacy team’s attention. As with any prescription, it was the pharmacist’s responsibility to confirm the match between the prescription and the condition by asking questions. The patients who received an antibiotic to treat or prevent cystitis were invited to participate in the study. Participation was free and voluntary. Verbal approval indicated consent to participate in the study. One or two minutes were required to complete the survey (Appendix A).

Community pharmacists were recruited through the Regional Union of Health Professionals (*Union Régionale des Profesionnels de Santé—URPS*). This organization sent an e-mail to all community pharmacists in the *Pays de la Loire* region of France. A virtual meeting was organized to present the study in detail and to explain the process. Sixteen community pharmacists agreed to participate in the study. This study aimed to assess compliance with the French guidelines, using the same modalities as the preliminary study.

### 2.3. Sample Size and Statistical Analysis

The percentage of compliance observed for the preliminary study (69%) was consistent with the literature. Grave et al. produced a similar study and obtained a compliance of 58% [19]. As a precaution, a margin was taken, and the theoretical percentage of compliance retained was 60%. Considering a theoretical proportion of 60%, an observed proportion of 69%, a type I error rate, α of 5%, and a power of 95%, the required sample size was 303 patients.

GraphPad^®^ (Prism 8.4.3 version) was used for the statistical analysis of the data [20]. The data were presented as frequency and percentages. The chi-square test and Fisher’s *t*-test were used to appreciate the association between the conformity and some factors. A *p*-value of <0.05 indicated a statistically significant difference, which became highly significant if the *p*-value was <0.001. R software (version 4.1.3) was also used to perform a logistic regression analysis [21]. A variable selection was determined to minimize the AIC, while focusing on known confounders. The absence of collinearity was also checked.

## 3. Results

A total of 375 forms were received, but 62 were not complete and 10 were rejected because the inclusion criteria were not verified (two women were pregnant and eight women were over the age of 65). In total, 303 forms were retained for analysis.

### 3.1. Patients’ Characteristics

The sample seems heterogenous, with a representativity of each age range but also of several profiles, including the risk factor and the first infection (Table 2). While 22.4% of women suffered from cystitis for the first time, nearly one in two (47.8%) had experienced at least one in the past 12 months. For 11.9% of them, it was even a recurrent cystitis. The risk of reporting a new episode within 3 months of the first one seemed to be verified; 31.7% of patients had at least one episode in the last 3 months. Most infections were classified as uncomplicated cystitis (79.2%) and some women were reported to be at risk of complication (8.9%).

Hygiene rules were explained for most of the women (56.4%). A urine culture was performed for 35.5% of patients, but a urine dipstick was performed for only 14.5% of them. Only a small proportion of patients received prescriptions containing an additional box of antibiotics (11.9%).

### 3.2. Prescriptions’ Characteristics

General practitioners were the prescribers in more than nine out of ten cases (Figure 1). Fosfomycin trometamol was the antibiotic prescribed for one in two women, while pivmecillinam was in second place (with 16.3% of prescriptions). Cefixime and ciprofloxacin shared third place (at 6.5%). The other antibiotics were anecdotic with less than 5% for each of them.

### 3.3. Compliance between Cystitis Management and Guidelines

The compliance of the prescriptions was assessed according to the valid recommendations at the time of the survey (Figure 2). A global average compliance of 61.4% was observed, with some disparities according to the type of cystitis. Approximately two-thirds of the cases of uncomplicated cystitis and recurrent cystitis followed the recommendations, whereas only 14.8% of the cases of cystitis at risk of complication did so. For the latter, a decrease in compliance was observed for each additional criterion (the compliance rate divided by two between the nature and the dosage, and then halved again with the addition of the duration).

To assess the association between some variables and the compliance of antibiotic therapy (nature, dosage, and duration), a logistic regression was performed (Figure 3). Ages between 56 and 65 years and having a prescription left with the office secretary increased the risk of non-compliance (with an OR of 2.6 and 4.52, respectively). Conversely, the absence of a urine culture slightly but significantly reduced non-compliance (with an OR between 0.58 and 0.99). Finally, the absence of a risk factor in the patient remains the criterion that most reduced non-compliance with antibiotic therapy (with an OR of 0.08).

A focus was placed on uncomplicated cystitis (n = 240). For this specific indication, the compliance between guidelines and antibiotic therapy was 66.3% (nature, dosage, and duration) (Figure 4). A dipstick test was performed in only 35 cases (14.6%), and a urine culture was performed twice as often (33.3%). Compliance with the antibiotic therapy and the urine examination led to a compliance rate of 5.8% (n = 14), ten times lower than compliance with the antibiotic prescription alone. Explanation of hygienic rules by the physician reduced this compliance to 2.9% (n = 7) (Figure 3).

## 4. Discussion

This study aimed to assess the compliance between prescriptions collected in community pharmacies and the French cystitis guidelines. An average compliance of 66% was observed. Two-thirds of cases of uncomplicated cystitis and cystitis at risk of complication were compliant for the antibiotic therapy, while only 15% of cases of recurrent cystitis were. These results appeared to be in line with the international literature. For example, even though Grave et al. conducted their study in a hospital, they reported similar compliance rates (58% and 61.4%, respectively) [19]. In this study, a similar finding was observed for the management of cystitis at risk of complication, with a compliance rate of 20% (14.8% in the study). The management of this type of infection seems to be less known to physicians. Several factors should be considered in determining whether antibiotic therapy is appropriate for the guidelines. The presence of risk factors, and therefore the type of cystitis, are factors that influence compliance with the recommendations, considering the type of urinary examination performed for uncomplicated cystitis decreased compliance with guidelines by a factor of 10.

The nature of the antibiotic appeared to be a major factor of compliance (Figure 2). For uncomplicated cystitis and recurrent cystitis, a quarter of the prescriptions were not in compliance with the nature of the microbial agent alone. Considering that almost 80% of cystitis cases were uncomplicated, it was expected that fosfomycin trometamol would be the first antibiotic prescribed (54.9%). However, it was even more surprising that quinolones were prescribed in 14.3% of cases, even though they are no longer indicated for this type of infection. In France, a focus on the good use of fluoroquinolones was published by the *SPILF* in 2015 [22]. This study highlights that they are very effective but must be prescribed sparingly to limit resistance and preserve the intestinal microbiota. Therefore, they should not be prescribed when another antibiotic can be used. Previous treatments must be considered (contraindication to prescribe a fluoroquinolone, if prescribed within the last 6 months), and the recommended dosage must be respected [23]. Similar studies have been conducted in other countries and found comparable difficulties in compliance with treatment guidelines [24,25,26,27,28]. In France, a clear decrease in lomefloxacin and norfloxacin prescriptions has been observed since they are no longer reimbursed [17]. Even if the recommendations might vary from one country to another, the fight against AMR is a major issue. The over-prescription of quinolones might lead to resistance and limited efficacy and should be restricted. In some countries, these antibiotics are prescribed in the majority of cases (51% of prescriptions in Houston, TX, USA) [25].

The nature of the antibiotic was not the only criteria considered. The addition of dosage to compliance had little impact on uncomplicated cystitis and recurrent cystitis, with a decrease in compliance of three points. A new decrease in compliance of six points was observed with the addition of the duration criteria (nature, dosage, and duration). The management of cystitis at risk of complication was a real problem, as each parameter reduced compliance with recommendations by half. Regarding the duration of antibiotic therapy, a Cochrane review by Lutters et al. reported the added value of short treatments [29]. They found that there was no significant difference in efficacy between short and long treatments, but that the latter caused more adverse effects. For this reason, many international recommendations tended to reduce the duration of antibiotic treatments used to limit AMR [30,31,32,33]. These recommendations should also be communicated to prescribers in the best possible way.

The next step is to bring the new recommendations to prescribers so that they may better manage these infections. A systematic meta-review from Francke et al. provided some answers [34]. The characteristics of healthcare providers, patients, and the environment must be considered. In addition, simple but varied strategies appeared to be more effective. In our case, the frequent updating of the recommendations seemed to be an obstacle to their application. It would be relevant to conduct the same study, with the new recommendations currently in effect, and to observe whether the reduction in the duration of the prescription of pivmecillinam has been retained.

According to Llor et al., the adherence of physicians to guidelines for the management of lower UTIs in women is poor [35]. It is reported that first-choice drug schedules were prescribed in only 17.5% of cases. A recent qualitative study was conducted to analyse primary-care provider decisions in the management of UTIs [36]. In this study, physicians were not familiar with fosfomycin trometamol, which explained its low prescription rate. The same problem was observed in France with pivmecillinam. Since 2015, in uncomplicated cystitis, pivmecillinam was really prescribed only 2–3 years later [17]. The authors noted that beliefs may lead healthcare providers to prescribe one antibiotic over another. For example, they believed that infection control was faster with fluoroquinolones than with trimethoprim-sulfamethoxazole or nitrofurantoin. They also added that the recommendations should be integrated into the medical software to ensure compliance.

Our study had several limitations. Even though we highlighted existing practice flaws, unfortunately our methodology did not allow us to help improve them. In addition, our study suffered from a collection and a recall bias because of the lack of contact between the pharmacist and the physician, as well as the inability to access the patient’s medical records. Indeed, the pharmacists had to collect all the information from the patients themselves. It can explain the collection of contradictory data in some rare cases (e.g., one episode was indicated in the current year, while the last episode was more than a year ago). Some questions must be approached with caution, especially those involving the patient’s history (such as previous infections, or previous treatment with a quinolone). Nevertheless, the use of the patient’s history via the business software or the pharmaceutical files (available with the patient’s health card in France), allowed us to limit this lack of information. Only pharmacists who volunteered to participate in the study were included, without randomization. Their motivation and the extension of the collection period provided the expected sample. Moreover, as it is impossible to be sure of the completeness of the reporting of UTI cases in each pharmacy during the study period, a patient-selection bias can be assumed. This bias seems limited by the proportion of each type of UTI collected; that is consistent with the epidemiology of this infection [3].

The pharmacist is the last line of defence before a patient takes a drug. Thus, the pharmacist must be aware of the recommendations in force and ensure that they are respected. If necessary, the pharmacist can contact the prescriber to adapt the prescription. As with physicians, it may be appropriate to assess pharmacists’ knowledge in this area. A study that assessed pharmacists’ knowledge of current recommendations would estimate the rate of prescribing that could be improved. This study provided an overview of compliance with antibiotic prescriptions for cystitis in women. Uncomplicated cystitis and recurrent cystitis benefitted from average management (concordance of approximately 66% between recommendations and prescriptions) but cystitis at risk of complication should be improved with a concordance rate of 15%. The management of UTIs also involved an examination. The example of simple cystitis showed us that urine cultures were performed instead of urine dipstick tests. Taking this criterion into account, in addition to antibiotic therapy, reduced the compliance rate for the management of uncomplicated cystitis to less than 10%.

Following this study, an intervention for prescribers should be considered to provide them with the tools to implement the new recommendations. The frequent updating of recommendations complicates their application. The availability of computer tools on the prescription software, mobile phone applications, or simple online tools might be one solution [37,38,39]. These tools can improve the knowledge translation from the guidelines to the physicians [40]. Interprofessional education, continuing professional development, or multifaceted, education-focused interventions are other proposals to improve adherence to recommendations [41,42]. A post-intervention analysis would then be appropriate to measure the effects of this intervention. Furthermore, because uncomplicated cystitis is such a common and simple UTI, its management by a pharmacist would be relevant. A dispensation under protocol by the pharmacist would allow them to answer this need of care, but also to respect the recommendations with the protocol used [43,44,45,46].

## Figures and Tables

**Figure 1 antibiotics-11-00976-f001:**
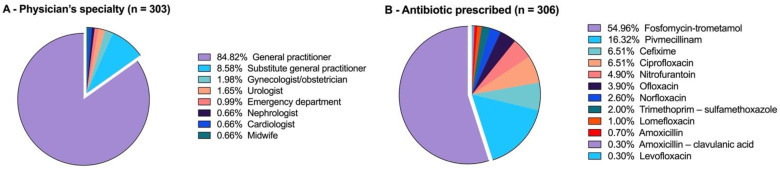
The origin of the prescriber (**A**) and the nature of the antibiotic prescribed (**B**) among the 303 selected prescriptions. In three cases, two antibiotics were prescribed at the same time for the same patient (n = 306).

**Figure 2 antibiotics-11-00976-f002:**
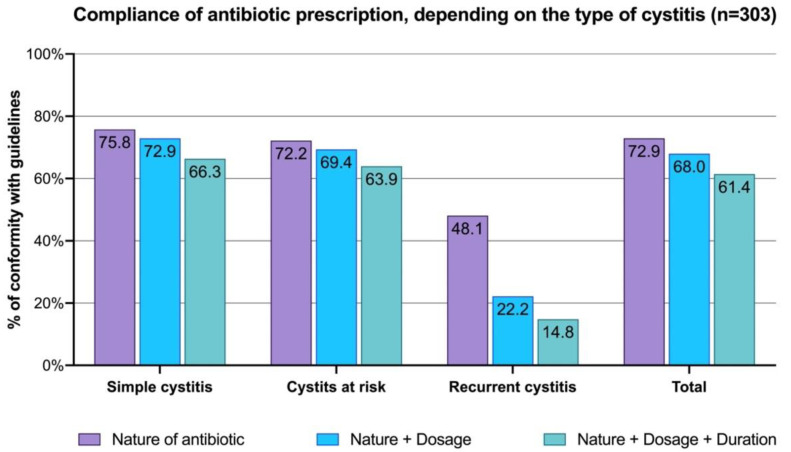
The compliance of antibiotic prescription, depending on the type of cystitis: uncomplicated cystitis, cystitis at risk of complication, and recurrent cystitis (n = 303). Compliance was examined based on the nature of the antibiotic, the nature and dosage of the antibiotic, or the nature, the dosage, and the duration of the antibiotic with the guidelines.

**Figure 3 antibiotics-11-00976-f003:**
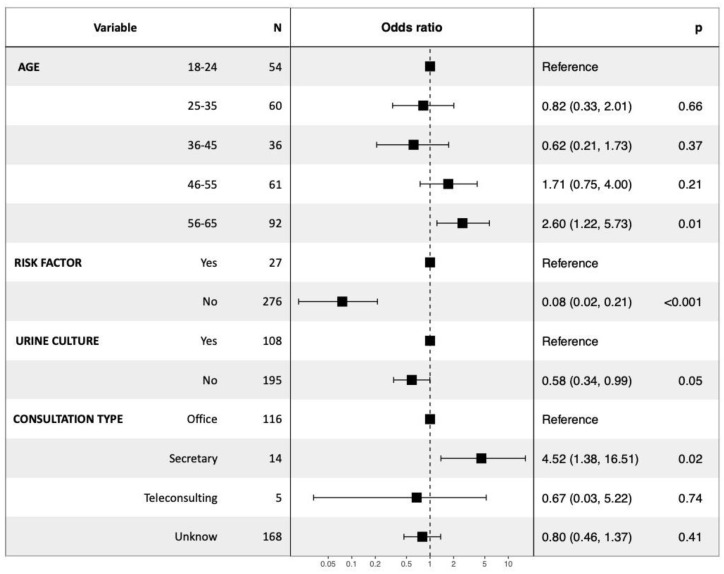
The impacts of the different factors which influenced the risk of non-compliance with antibiotic therapy for cystitis. Logistic regression was conducted with an α risk of 5%. A *p*-value of <0.05 was considered significant, a *p*-value of <0.01 was considered very significant, and a *p*-value of <0.001 was considered highly significant.

**Figure 4 antibiotics-11-00976-f004:**
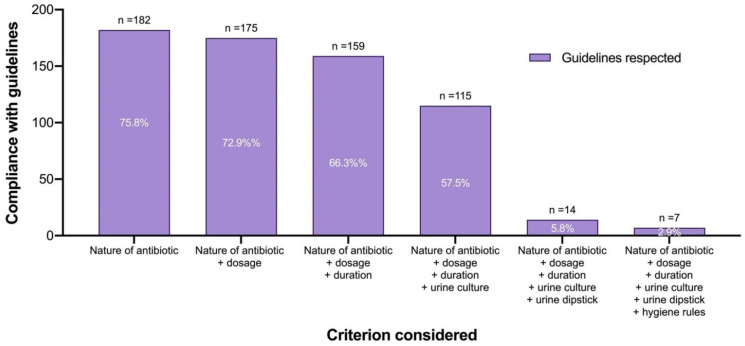
The compliance of uncomplicated cystitis management with recommendations using cumulative criteria (n = 240). Each column represented a new compliance criterion. The starting point was the compliance of the nature of the antibiotic, then its dosage, and its duration. Then, the compliance of the urine examination, urine culture, and urine dipstick were added. Finally, the explanation of hygiene rules was included.

**Table 1 antibiotics-11-00976-t001:** French guidelines for cystitis management *, according to the type of cystitis, during the preliminary study and the study.

Uncomplicated Cystitis	Cystitis at Risk of Complication	Recurrent Cystitis
Fosfomycin trometamol 3 g, single dose Pivmecillinam 400 mg, twice daily for 5 days	**If the result of urine culture is known:** Amoxicillin 1 g, three daily for 7 days Pivmecillinam 400 mg, twice daily for 7 days Nitrofurantoin 100 mg, three daily for 7 days Fosfomycin trometamol 3 g, three doses each administered 48 h apart (D1-D3-D5) Trimethoprim 150 mg, once daily for 5 days	**Curative treatment** (<1 episode per month) Fosfomycin trometamol 3 g, single dose Pivmecillinam 400 mg, twice daily for 5 days
	**If the treatment cannot be delayed** Nitrofurantoin, 7 days Fosfomycin trometamol 3 g, three doses each administered 48 h apart (D1-D3-D5)	**Prophylactic treatment** (>1 episode per month) - Trimethoprim 150 mg, once daily - Trimethoprim-sulfamethoxazole 80/400 mg, once daily - Fosfomycin trometamol 3 g, once weekly

* Recommendations used as a reference; published by the French Infectious Diseases Society in 2018 [10].

**Table 2 antibiotics-11-00976-t002:** Patients’ characteristics (n = 303).

Patients’ Characteristics		Number of Patients (%)
Age	18–24	54 (17.8)
	25–35	60 (19.8)
	36–45	36 (11.9)
	46–55	61 (20.1)
	56–65	92 (30.4)
Type of cystitis	Uncomplicated	240 (79.2)
	Recurrent	36 (11.9)
	At risk of complication	27 (8.9)
Risk factor	Abnormalities of the urinary tract	18 (5.9)
	Severe immunodeficiency	9 (3.0)
	No	276 (91.1)
First infection	Yes	68 (22.4)
	No	235 (77.6)
Last infection	<15 days	13 (4.3)
	<1 month	26 (8.6)
	<3 months	57 (18.8)
	<12 months	70 (23.1)
	>12 months	58 (19.1)
	Do not know	11 (3.6)
	NA	68 (22.4)
Episode frequency in the last 12 months	0	128 (42.2)
	1	55 (18.2)
	2	47 (15.5)
	3	37 (12.2)
	4	18 (5.9)
	>4	18 (5.9)
Quinolones in the last 6 months	Yes	15 (5.0)
	No	287 (94.7)
	Do not know	1 (0.3)
Supplementary box	Yes	36 (11.9)
	No	267 (88.1)
Urine dipstick realized	Yes	44 (14.5)
	No	259 (85.5)
Urine culture realized	Yes	108 (35.6)
	No	195 (64.4)
Hygienic and dietary rules explained	Yes	171 (56.4)
	No	132 (43.6)

## Data Availability

The data presented in this study are available on request from the corresponding author.

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
