# Peer review of "Assessment of the Compliance of Cystitis Management According to French Recommendations through the Analysis of Prescriptions Collected in Community Pharmacies"

_antibiotics, 2022, doi:10.3390/antibiotics11070976_

Round 1

Reviewer 1 Report

I have read with interest the manuscript by Piraux et al., since AMR represents a significant concern.

I have some comments to be addressed to improve the manuscript:

The latest urological guideline defines recurrent UTIs as " Recurrent UTIs (rUTIs) are recurrences of uncomplicated and/or complicated UTIs, with a frequency of at least three UTIs/year or two UTIs in the last six months."

more information at: https://d56bochluxqnz.cloudfront.net/documents/full-guideline/EAU-Guidelines-on-Urological-Infections-2022.pdf

- once you described an abbreviation, please use the abbreviated form (Ex. E. coli)

- please correct: "a p-value of < 0.01 was considered very significant and a p-value of < 0.01 was considered highly significant." (line 191)

- I suggest the review of the article by a native English speaker. Some phrases are long and hard to understand, while some conjugations are not properly realized. 

- please clarify:

56 forms were collected but 52 retained (1 pregnant patient, 1 patient older than 65 years, and 2 incomplete forms). 41 prescriptions were considered compliant, representing an observed compliance percentage of 69.2%

41 prescriptions out of 52 forms would be a higher percentage of compliance.

Author Response

Thank you for your thoughtful and relevant comments, they helped us improve the manuscript. Every remark has been taken into account to facilitate understanding. We thank you for the careful reading that allowed us to correct typing errors.

The latest urological guideline defines recurrent UTIs as " Recurrent UTIs (rUTIs) are recurrences of uncomplicated and/or complicated UTIs, with a frequency of at least three UTIs/year or two UTIs in the last six months."

more information at: https://d56bochluxqnz.cloudfront.net/documents/full-guideline/EAU-Guidelines-on-Urological-Infections-2022.pdf

âž” As specified in the article (line 103), we used the French recommendations in effect at the time of the study. In light of these new recommendations, we clarified the nuance of the definition of recurrent cystitis in the study (line 46).

- once you described an abbreviation, please use the abbreviated form (Ex. E. coli)

âž” We have replaced the terms with the appropriate abbreviations.

- please correct: "a p-value of < 0.01 was considered very significant and a p-value of < 0.01 was considered highly significant." (line 191)

âž” Our apologies for this error in the proofreading of the legend, thank you for your vigilance.

- I suggest the review of the article by a native English speaker. Some phrases are long and hard to understand, while some conjugations are not properly realized. 

âž” Editorial assistance was provided by Speak the Speech Consulting. However, we have made modifications to simply come sentences. We hope that the manuscript will be more understandable.

- please clarify:

56 forms were collected but 52 retained (1 pregnant patient, 1 patient older than 65 years, and 2 incomplete forms). 41 prescriptions were considered compliant, representing an observed compliance percentage of 69.2%

41 prescriptions out of 52 forms would be a higher percentage of compliance.

âž” Once again, thank you for your careful reading. We have clarified this result (41/52 = nature of antibiotic OK – 36/52 = nature + dosage + duration OK).

Reviewer 2 Report

The manuscript assesses the compliance of cystitis management according to French recommendations through the analysis of prescriptions collected in community pharmacies. If we consider the problem of antibiotic resistance, a similar study is interesting and can explain why this resistance is growing and their risk factors. From this study it is clear that two thirds of the cases of uncomplicated cystitis and recurrent cystitis followed the French recommendations. These data represent a real alarm in terms of drug resistance if we consider that only 14.8% of cases of cystitis at risk of complication follow the recommendations. The authors suggest that the pharmacist is the last line of defense before a patient takes a drug; therefore, for this reason, the pharmacist must be aware of the recommendations in force. According to this point of view, a question spontaneously arises: did the authors evaluate the pharmacist's knowledge of the current recommendations with the help of questionnaires? Then, the manuscript is well written. However, please the authors modify the sentence on line 134 as follows: “A p-value of 134 <.05 indicates a statistically significant difference, which becomes highly significant if p-value <135 .001. Software R (version 4.1.3) was also used to perform logistic regression analysis "(the original sentence was" A p-value of 134 <.05 was considered a statistically significant and highly significant difference if p-value <135. 001. R software (version 4.1.3) was also used to perform logistic regression analysis).

Author Response

We thank you for your careful reading of our study and the suggestions you made. We particularly appreciate the value of conducting a new study that focuses on the pharmacist’s knowledge of the new recommendations.

The manuscript assesses the compliance of cystitis management according to French recommendations through the analysis of prescriptions collected in community pharmacies. If we consider the problem of antibiotic resistance, a similar study is interesting and can explain why this resistance is growing and their risk factors. From this study it is clear that two thirds of the cases of uncomplicated cystitis and recurrent cystitis followed the French recommendations. These data represent a real alarm in terms of drug resistance if we consider that only 14.8% of cases of cystitis at risk of complication follow the recommendations. The authors suggest that the pharmacist is the last line of defense before a patient takes a drug; therefore, for this reason, the pharmacist must be aware of the recommendations in force. According to this point of view, a question spontaneously arises: did the authors evaluate the pharmacist's knowledge of the current recommendations with the help of questionnaires?

âž” The pharmacist's knowledge of the current recommendations has not been conducted, but it would be very relevant. Indeed, it would allow us to assess the rate of pharmacists who are aware of the latest recommendations, which are constantly changing. It could also provide information on the training and information channels used. Finally, it would be interesting to know if the physician adapts his prescription after an exchange with the pharmacist (in the case of a prescription not in accordance with the recommendations). We will try to develop such a study in the future if the opportunities arise. This thought was completed in the discussion (line 290).

Then, the manuscript is well written. However, please the authors modify the sentence on line 134 as follows:

“A p-value of 134 <.05 indicates a statistically significant difference, which becomes highly significant if p-value <135 .001. Software R (version 4.1.3) was also used to perform logistic regression analysis "(the original sentence was"

A p-value of 134 <.05 was considered a statistically significant and highly significant difference if p-value <135. 001. R software (version 4.1.3) was also used to perform logistic regression analysis).

âž” We have replaced with the proposed suggestion.